# Correlations Between Gender, Age, and Occupational Factors on the Work Ability Index Among Healthcare Professionals

**DOI:** 10.3390/healthcare13070702

**Published:** 2025-03-22

**Authors:** Valerio Paneni, Cristiana Gambelunghe, Luca Tomassini, Giuliana Buresti, Bruna Maria Rondinone, Benedetta Persechino, Daniela Fruttini, Marco dell’Omo, Chiara Pucci, Angela Gambelunghe

**Affiliations:** 1Occupational Medicine, Respiratory Diseases and Toxicology Section, Department of Medicine and Surgery, University of Perugia, 06129 Perugia, Italy; paneni.valerio@gmail.com (V.P.); marco.dellomo@unipg.it (M.d.); chiarapucci9@gmail.com (C.P.); 2Forensic Medicine, Forensic Science and Sports Medicine Section, Department of Medicine and Surgery, University of Perugia, 06129 Perugia, Italy; cristiana.gambelunghe@unipg.it; 3School of Advanced Studies, University of Camerino, 62032 Camerino, Italy; luca.tomassini@unicam.it; 4Department of Occupational and Environmental Medicine, Epidemiology and Hygiene, Italian Workers’ Compensation Authority (INAIL), Monte Porzio Catone, 00144 Rome, Italy; g.buresti@inail.it (G.B.); b.rondinone@inail.it (B.M.R.); b.persechino@inail.it (B.P.); 5Department of Medicine and Surgery, University of Perugia, 06129 Perugia, Italy; daniela.fruttini@unipg.it

**Keywords:** Work Ability Index, healthcare workers, aging work population, occupational health, gender differences

## Abstract

**Background**: The Work Ability Index (WAI) measures how well employees’ abilities match their job demands. This study assessed the WAI among health workers and explored how age, gender, and job roles affected it. The research was conducted in a central Italian hospital, with a focus on health workers undergoing health surveillance between September 2020 and April 2021. **Methods**: Data were collected using validated questionnaires that assessed the WAI and risk factors for metabolic syndrome among participants. Demographic information, including age, gender, and occupation, was also obtained. The study involved 1847 health workers, with an average age of 43 years, predominantly women (67.6%). Occupational categories included administrative staff, nurses/healthcare workers (HCWs)/midwives, physicians, and healthcare technicians. Statistical analyses, such as *t*-tests, ANOVA, and chi-squared tests, were performed to explore the relationships between WAI scores and demographic/occupational variables. **Results**: The study suggested a relationship between WAI scores and gender, age, and occupation. Men workers exhibited higher mean WAI scores than women workers, while older workers (>55 years) had lower WAI scores compared with their younger counterparts. WAI scores varied by job role, with physicians scoring the highest. **Conclusions**: The findings suggested that demographic and occupational factors were associated with variations in work ability among health workers. These findings can help improve workforce management, occupational health, and research on aging workers. However, it is important to acknowledge the limitations of this study. Given its cross-sectional design, causal inferences cannot be established, and further longitudinal research is needed to confirm these findings and explore potential causal relationships.

## 1. Introduction

The concept of work ability has become increasingly relevant in modern workplaces, particularly in light of demographic shifts and an aging workforce. First introduced in 1981, the Work Ability Index (WAI) was developed as a tool to assess how well employees could meet occupational demands given their psychophysical capabilities [1,2,3]. As populations age globally, the proportion of older workers in the labor force is growing, bringing new challenges and considerations for workforce sustainability [4].

Aging affects both physical and cognitive functions, altering employees’ abilities to perform their job roles effectively. With age, individuals experience physiological changes such as reduced muscle strength, cardiovascular efficiency, and lung function, alongside cognitive shifts that impact memory, processing speed, and problem-solving abilities. These changes inevitably influence how employees approach their work and adapt to occupational demands [5,6,7].

In healthcare, where physical and mental demands are significant, understanding the factors that influence work ability is crucial [8,9]. The present study aimed to evaluate the Work Ability Index among healthcare professionals and examine how various factors—both non-modifiable (such as age and gender) and modifiable (such as job type)—affected work ability. By exploring these correlations, this research sought to provide insights that could help optimize workforce management, improve job sustainability, and ensure that healthcare professionals maintain a high level of work ability despite demographic shifts.

### 1.1. Population Aging

Over the past decades, global population aging has accelerated due to declining birth rates and increased life expectancy. The number of individuals aged 60 and older is projected to double by 2050 [10]. In Europe, the percentage of people over 65 rose from 17.1% in 2008 to 19.7% in 2018, with a median age increase from 40.4 years in 2008 to 44 years in 2020 and an expected rise to 49 years by 2070 [11]. In Italy, this trend is particularly pronounced, with the over-65 population increasing from 9.3% in 1960 to 22.8% in 2019 and projections suggesting it could reach 33% by 2040–2045 and 31–37% by 2060 [12,13]. This demographic transformation, driven by the post-World War II baby boom and persistently low birth rates since the 1980s, has significant implications for labor markets and economic structures [14].

### 1.2. Effect of Aging

Aging is a continuous and multifaceted process that brings about gradual changes in both physical and cognitive functions, typically beginning in the fourth decade of life. Physiologically, aging leads to a decline in muscle strength [9,15] and an increased risk of cardiovascular diseases [16], accompanied by functional changes such as reduced heart rate variability and a lower maximum heart rate [17]. Lung function also deteriorates over time, following an initial phase of development that peaks around age 20 in women and 25 in men. From approximately age 35 onward, both structural and functional lung capacity begin to decline [18,19].

Another key aspect of aging is the progressive reduction in maximal oxygen consumption (VO_2max_), which starts once full physical maturity is reached, typically by age 30 [14]. Additionally, aging is closely associated with metabolic syndrome, a cluster of conditions that significantly increase the risk of chronic diseases [20].

Cognitive function is also affected by aging but in distinct ways. “Crystallized” intelligence, which includes accumulated knowledge and acquired skills, generally remains stable. In contrast, “fluid” intelligence, responsible for problem-solving, working memory, and processing speed, tends to decline with age [21]. Understanding these changes is essential for promoting healthy aging and developing strategies to support both physical and cognitive well-being.

### 1.3. Aging and Work

As workforce demographics shift, the average age of workers continues to rise [22]. The process of information processing in work environments can be divided into three key stages:The sensory-perceptive system, responsible for receiving information from the environment through the senses.The cognitive system, which processes sensory input and interacts with memory.The motor system, which executes actions based on cognitive decisions.

Each of these systems slows down with age, potentially affecting work performance [14]. However, despite the physical and cognitive decline, older workers often compensate by leveraging their extensive experience and accumulated knowledge. Research suggests that while older employees may have lower physical and mental endurance compared with younger workers [23], they are less prone to burnout, which is more commonly seen in younger employees at the start of their careers [24].

Workplace values and priorities also evolve with age. Older workers tend to place a higher emphasis on responsibility and the significance of their roles, whereas younger employees prioritize learning and career growth. Meanwhile, middle-aged workers often balance work commitments with family responsibilities [25].

One challenge associated with an aging workforce is skill obsolescence. Some studies indicate that older employees may struggle with acquiring new skills, adapting to technological advancements, and undergoing training for updated work practices [25]. However, few organizations recognize aging as an opportunity. Actively investing in aging workers and leveraging their experience could be a strategic approach to addressing current labor shortages [26].

### 1.4. Work Ability Index

The concept of work ability was first introduced in 1981 by Ilmarinen J. et al. and was defined as the answer to the question, “*How good are workers at present and in the near future and how able are they to do their job with respect to work demands, health, and mental resources?*” [27].

The “Work Ability Index” was created by the authors to measure the level of employees’ work ability, a score measured using a questionnaire which was divided into seven areas:*Subjective estimation of present work ability compared with the lifetime best* (1–10 points);*Subjective work ability in relation to both physical and mental demands of the work* (2–10 points);*Number of diagnosed diseases* (1–6 points);*Subjective estimation of work impairment due to disease* (1–6 points);*Sickness absence during past year* (1–6 points);*Own prognosis of work ability after two years* (1–7 points);*Psychological resources (enjoying daily tasks, activity and life spirit, optimistic about the future)* (1–4 points).

The results can therefore vary between 7 and 49 points; based on the score obtained, the work ability is divided into four classes: poor, moderate, good, and excellent.

The WAI can be used to define the risk of early retirement for illness [28] or prolonged absence for illness [29,30]. WAI can be used to find links between work skills and physical and psychosocial risk factors in the work environment, thus enabling work to improve the longevity of the workforce [31].

In a study conducted among nurses working in intensive care and the emergency department, the correlation between WAI and cognitive failures was observed, indicating that a decrease in WAI corresponds to an increase in cognitive failures [32].

### 1.5. WAI and Non-Modifiable Variables (Gender and Age)

The relationship between age and WAI scores has been widely studied, with inconsistent findings. Some studies report a negative correlation between age and work ability [32,33,34,35,36], while others find no significant association between the two variables [37,38,39].

Similarly, the impact of gender on WAI scores remains debated. Several studies report no significant difference in work ability between men and women [37,38,39], although some evidence suggests that men tend to rate their own work ability higher than women do [38]. Other studies highlight gender-based differences in mean WAI scores [35,36] or variations based on job type. Specifically, women working in physically demanding or mixed (physical and mental) jobs tend to report higher WAI scores than men, whereas in mentally focused jobs, men score higher than women [40].

Nilsson et al. further explored this gender disparity, emphasizing that high physical workload among women, particularly in social and healthcare sectors, contributes to these differences [41].

This study sought to bridge existing gaps in research by examining how gender, age, and occupational factors influenced work ability among healthcare professionals. Rather than simply comparing Work Ability Index (WAI) scores between men and women, our analysis delved deeper into the occupational challenges and physiological differences that may contribute to these disparities.

Additionally, we explored the impact of aging on work ability, with a particular focus on the healthcare sector, where both physical and cognitive demands are substantial. As workers grow older, their ability to meet job requirements may change, making it essential to understand how aging interacts with different professional roles.

Another key objective of this research was to investigate work ability across various healthcare professions, determining whether certain groups—such as nurses, physicians, or healthcare technicians—were more susceptible to a decline in work ability and identifying the underlying reasons for these differences.

By addressing these aspects, this study enhances the existing body of literature by offering a more comprehensive perspective on the factors influencing work ability in healthcare. Furthermore, it underscores the necessity of targeted occupational health strategies, tailored to mitigate gender disparities, address aging-related challenges, and consider the specific demands of different healthcare roles. These insights are crucial for fostering a sustainable and resilient workforce in the healthcare sector.

#### Scope and Research Question

This study aimed to explore the relationships between gender, age, and occupational roles in determining work ability among healthcare professionals. Specifically, we examined how gender differences influenced WAI scores; whether older healthcare workers exhibited significantly lower work ability compared with their younger colleagues; and how different occupational roles, such as physicians, nurses, and healthcare technicians, contributed to variations in WAI scores. By addressing these aspects, this study sought to answer the research question, “How do gender, age, and occupational roles influence the Work Ability Index (WAI) among healthcare professionals in a hospital setting?”. Furthermore, we investigated whether gender- and age-related disparities in WAI were mediated by occupational roles, shedding light on the specific challenges faced by different professional categories within the healthcare sector.

## 2. Materials and Methods

### 2.1. Design of the Study

A cross-sectional study was conducted on health workers in a hospital in central Italy through the administration of two questionnaires during health surveillance visits (Legislative decree 81/08). The first questionnaire was validated for the calculation of the work ability index, and the second investigated the risk factors of metabolic syndrome (BMI, waist circumference, total cholesterol, LDH, LDL, triglycerides, glycemia, uricemia, systolic blood pressure, diastolic blood pressure, and smoking habit). A personal data sheet was attached to the two questionnaires to determine the age, gender, and job.

### 2.2. Population of the Study

The study population included health workers under surveillance for Legislative Decree 81/2008 in the period from September 2020 to April 2021; workers accepted and signed the privacy consent according to Law 675/1996. A total of 1847 workers were considered, 599 men and 1248 women, and the average age was 43 years. The study population consisted of 207 administrative workers, 882 non-medical health workers (nurses, socio-health workers, midwives), 544 physicians, and 214 health technicians. Considering the WHO definition of an older adult worker, the study population was made up of 474 older adult workers (≥55 years of age) and 1372 young workers (<55 years of age). The study population was divided into 4 categories: administrative, physicians (this included both physicians in specialty training and medical managers), nursing and health care (nurses, HCWs, and midwives), and health care technicians (all hospital staff who did not fall into the previous categories). The study protocol was approved by the Ethics Committee of Umbria Region (CET) (Approval No. 4809/24). The research was conducted in accordance with the Declaration of Helsinki. All participants provided informed consent and voluntarily participated.

#### Inclusion and Exclusion Criteria

The study included healthcare professionals undergoing occupational health surveillance at a hospital in central Italy between September 2020 and April 2021, in compliance with Legislative Decree 81/08. The inclusion criteria were as follows:Being a healthcare professional (nurses, physicians, healthcare technicians, and midwives) employed at the facility under study.Undergoing mandatory occupational health surveillance.Providing informed consent in accordance with Law 675/1996.

The following individuals were excluded from the study:Workers not belonging to healthcare categories (e.g., administrative staff not involved in clinical activities and non-healthcare support personnel).Individuals who did not provide informed consent.Individuals not subject to mandatory occupational health surveillance during the study period.

### 2.3. Data Collection Tools

The study utilized three main instruments for data collection. The first was the Work Ability Index (WAI) questionnaire, a validated tool designed to assess employees’ perceived ability to perform their job-related tasks. This questionnaire provided insights into how well individuals felt they could meet the demands of their profession. The Work Ability Index (WAI) is a tool developed to assess an individual’s work ability in relation to job demands, health status, and personal resources. It consists of seven dimensions evaluating key aspects such as current work ability, diagnosed diseases, functional limitations, sick leave, self-perceived future work ability, and mental resources like concentration and motivation. The WAI score ranges from 7 to 49 and classifies work ability into four categories: poor, moderate, good, and excellent [27].

Subsequently, we cross-referenced workers’ demographic data with anthropometric and laboratory data (age, gender, BMI, waist circumference, total cholesterol levels, LDL and HDL cholesterol, triglycerides, glycemia, and uricemia, as well as systolic and diastolic blood pressure) collected during occupational health surveillance visits, in accordance with Legislative Decree 81/08.

### 2.4. Statistical Analysis

The quantitative data collected are presented with mean and standard deviation for the groups of patients analyzed. Groups were formed according to certain sample characteristics. Qualitative variables were described in percentages. In some cases, data were represented graphically. For quantitative data, the Kolmogorov–Smirnov test was used to determine whether the data were normally distributed. Based on the result obtained, the non-parametric Mann–Whitney test was used for comparison between two groups. For comparison between multiple groups (above two) we used the Kruskal–Wallis test. Post hoc analysis used the Bonferroni or the Bonferroni–Dunn test. A Chi-Square test was used in the analysis of qualitative contingency tables. A *p*-value below 0.05 was considered statistically significant. The processing was performed with SPSS 25 software for Windows (IBM Corp., Armonk, NY, USA).

## 3. Results

A total of 1847 workers were considered; 32.4% of the participants were men, while 67.6% were women. Regarding age distribution, 74.3% of the workers were under the age of 55, while 25.7% were 55 or older. When considering occupation, 11.2% of participants were classified as administrative staff; 47.8% were nurses, healthcare workers (HCWs), or midwives; 29.5% were physicians; and 11.6% were technicians (Table 1). The mean WAI within the women working population was 41.22 ± 5.5 with a median and interquartile range (IQR) of 42.0 ± 7.0. Among men, the mean WAI was 42.50 ± 5.1 with a median and IQR of 43.50 ± 5.50, indicating a statistically significant difference between genders (*p* < 0.001). The older working population (≥55 years old) exhibited an average WAI of 39.51 ± 5.9 with a median and IQR of 40.00 ± 7.50, in contrast to the younger working population, which averaged 42.37 ± 5.0 with a median and IQR of 43.00 ± 5.50; this disparity was also deemed statistically significant (*p* < 0.001).

Considering WAI across various occupations, we found that physicians possessed the highest WAI with an average of 43.30 ± 4.08 (median ± IQR: 44.00 ± 5.00), health care technicians had an average WAI of 42.36 ± 4.93 (median ± IQR: 44.00 ± 6.00), and administrative staff had an average WAI of 41.51 ± 5.30 (median ± IQR: 42.40 ± 6.00). Finally, nurses/HCWs/midwives had an average WAI of 40.47 ± 5.91 (median ± IQR: 42.00 ± 6.75). These discrepancies were statistically significant (*p* < 0.001) (Figure 1).

The mean, SD, and confidence intervals (CIs) of WAI across activities revealed significant differences between the administrative and medical groups, between the nurses/HCWs/midwives group and the medical and health technician groups, and between the medical group and the administrative and nurses/HCWs/midwives groups, as well as between the health technician group and the nurses/HCWs/midwives group.

In this study, the average WAI related to workers’ professions was computed, accounting for the gender-based segregation of the entire worker cohort. Within the working population of women, the Kruskal–Wallis test demonstrated statistical significance differences between professions (*p* < 0.001). The highest average WAI was observed within the cohort of physicians with a mean value of 43.30 ± 3.68 and a median and IQR of 44.00 ± 5.00. In descending order, we found health technicians with a mean WAI of 42.04 ± 5.04 and a median and IQR of 43.00 ± 6.00 and administrative staff with a mean WAI of 40.82 ± 5.83 and a median and IQR of 42.00 ± 7.00. Finally, the nurses/HCWs/midwives group had a mean value of 40.10 ± 5.90 and a median and IQR of 42.00 ± 7.00. In the working population of men, statistical significance was also found (*p* < 0.001), despite being less pronounced compared with the working population of women. The highest average WAI was found within the group of physicians that reached a mean value of 43.16 ± 4.57 and a median and IQR of 44.00 ± 5.00, followed, in descending order, by health technicians with a mean value of 42.97 ± 4.69 and a median and IQR of 44.00 ± 5.00 and administrative staff with 42.54 ± 4.18 and 43.30 ± 4.25, respectively. Finally, the nurses/HCWs/midwives group reached a mean value of 41.61 ± 5.82 and 43.00 ± 6.50 (Figure 2).

The statistical significance of the difference in WAI for the same profession between women and men workers was confirmed solely within the nurses/HCWs/midwives group (*p* < 0.001). The Bonferroni test revealed a statistically significant difference only within the medical nurses/HCWs/midwives group.

Moreover, the average WAI related to profession was examined, with the working population divided by age into two groups: younger workers and older workers. Among younger workers, there existed statistical significance between the task and the WAI according to the Kruskal–Wallis test (*p* < 0.001). The highest WAI was observed within the administrative sector with a mean value of 43.76 ± 3.85 and a median and IQR of 44.25 ± 5.63. In descending order, we found physicians with a mean value of 43.63 ± 3.88 and a median and IQR of 44.00 ± 4.00, followed by health technicians with a mean value of 43.44 ± 3.84 and a median and IQR of 44.00 ± 5.00. Finally, the nurses/HCWs/midwives group reached a mean WAI of 41.61 ± 5.82 and a median and IQR of 42.50 ± 7.00. Conversely, among older workers, the highest WAI was found within the physicians’ cohort with a mean of 41.62 ± 6.67 and a median and IQR of 41.75 ± 6.00. In descending order, we found health technicians with a mean WAI of 40.32 ± 6.03 and a median and IQR of 41.00 ± 8.00 and administrative staff with a mean WAI of 39.85 ± 5.62 and a median and IQR of 41.00 ± 7.00. Finally, we found the nurses/HCWs/midwives group with a mean WAI of 37.96 ± 6.18 and a median and IQR of 39.00 ± 8.25.

The statistical significance of the difference in WAI, for the same profession, between younger and older workers was confirmed by the Mann–Whitney U test for all four tasks. Bonferroni’s test indicated statistical significance in the difference in WAI, for the same task, between men and women in all four groups (Figure 3).

Table 2 presents the values of the WAI and the seven dimensions categorized by gender and age class.

The results indicated that total WAI, work capacity in relation to demands (W2), number of pathologies diagnosed by the physician (W3), and psychological resources (W7) tended to be higher in men compared with women (*p* < 0.05), except for the current work capacity (W1) dimension. For dimensions expressed by categorical variables, such as a reduction in work capacity due to illness (W4), absences due to illness in the last 12 months (W5), and personal forecast of work capacity in the following two years (W6), p-values of the Chi Square test were almost consistently significant, with lower percentages of positive situations observed in women, except for the W5 dimension “absences due to illness”, where there were no statistically significant association with gender.

Regarding age, the total WAI tended to decrease significantly with age (*p* < 0.001). The W3 dimension “pathologies diagnosed by a physician” (*p* < 0.001) and W7 dimension “psychological resources” (*p* = 0.002) tended to be higher among younger individuals. Concerning the dimensions related to the reduction in work capacity due to illness (W4), absences due to illness in the last 12 months (W5), and personal forecast of work capacity in the following two years (W6), *p*-values of the Chi-Square test consistently indicated significance, with lower percentages of positive situations observed among the older group of workers.

Table 3 displays the results of the total WAI and the seven dimensions categorized by different professions, classified into the four classes: administrative, nurses/HCWs/midwives, physicians, and healthcare technicians. In the W1 and W2 dimensions (current work ability and work capacity in relation to demands, respectively) nurses/HCWs/midwives consistently showed lower values compared with administrative and technicians. In the W3 dimension (number of pathologies diagnosed by the physician), physicians had the highest values among all job categories. Psychological condition values (W7) were notably lower among nurses/HCWs/midwives compared with physicians and technicians. Post hoc multiple comparisons were conducted using the T3 Dunnett test and Bonferroni.

For dimensions W4 (reduction in work capacity due to illness), W5 (absences due to illness in the last 12 months), and W6 (personal forecast of work capacity in the following two years), p-values of the Chi Square test consistently indicated significance, with lower percentages of positive situations for nurses/HCWs/midwives compared with physicians.

## 4. Discussion

Numerous studies consistently demonstrate an inverse relationship between age and the Work Ability Index (WAI) [27,35,41,42]. The present study corroborated this trend, indicating that older healthcare professionals exhibited lower WAI scores than their younger counterparts. However, the existing literature provides mixed results regarding the magnitude and causes of this decline. For example, some studies attribute the reduction in WAI to age-related physiological deterioration, including diminished muscle strength, cardiovascular efficiency, and cognitive function [9,15,17,20]. Others suggest that workplace factors, such as excessive workloads, lack of ergonomic adaptations, and inadequate job flexibility, significantly contribute to the observed decline in work ability among older employees [43,44]. The present findings supported the latter view, highlighting the importance of occupational factors in determining work ability outcomes.

The study also found significant gender disparities in WAI scores, with women reporting lower values than men. This result aligned with prior research indicating that women healthcare workers experienced higher physical and psychological workloads due to job demands and caregiving responsibilities outside the workplace [35,36,38]. One possible explanation was that women in healthcare professions, particularly in nursing and caregiving roles, were more likely to experience high physical strain and burnout, leading to a perceived decline in work ability [41]. Additionally, gender-based differences in health conditions and access to workplace support may contribute to this disparity. Studies suggest that women workers are more prone to musculoskeletal disorders and chronic stress, which could negatively impact their WAI scores [37,38]. However, conflicting evidence exists as some studies report no significant gender differences in WAI, emphasizing the need for further research on gender-specific occupational risks [39].

Occupational roles emerged as another significant determinant of WAI, with nurses, healthcare workers (HCWs), and midwives reporting the lowest scores compared with physicians and health technicians. This finding was consistent with previous studies highlighting the physically and emotionally demanding nature of nursing and caregiving professions [32,33,34]. Nurses and HCWs frequently work long shifts, engage in physically strenuous tasks, and experience high emotional exhaustion due to patient care responsibilities. As a result, they are at greater risk of work-related fatigue, burnout, and decreased work ability over time [42]. In contrast, physicians and technicians generally report higher WAI scores, likely due to differences in job demands, work schedules, and task autonomy. Physicians, for example, often have more control over their schedules and may experience lower levels of physical exertion compared with nurses and HCWs.

One of the key contributions of this study was the in-depth analysis of WAI dimensions across different demographic and occupational groups. The results indicated that psychological resources played a critical role in work ability, with younger workers and men employees reporting higher levels of optimism and job engagement than their older and women counterparts. This aligned with the existing literature suggesting that psychological factors, such as motivation, job satisfaction, and perceived work control, influenced work ability scores [21,44]. Given the decline in psychological resources with age, interventions targeting mental well-being and workplace support systems could help mitigate the negative effects of aging on work ability.

The study’s findings have important implications for workforce management and occupational health interventions. Given the observed decline in WAI with age, healthcare institutions should implement targeted strategies to support older workers. These may include flexible work arrangements, ergonomic workplace adaptations, and job redesign initiatives that accommodate age-related physical and cognitive changes. Moreover, addressing gender disparities in work ability requires policies that promote work–life balance, ensure equitable access to health resources, and mitigate occupational stressors disproportionately affecting women workers.

Future research should focus on longitudinal studies to establish causal relationships between demographic factors and work ability trends over time. Additionally, further exploration of organizational policies, psychosocial work conditions, and job satisfaction factors could provide deeper insights into the mechanisms influencing WAI. Investigating the role of workplace interventions, such as mentorship programs and occupational health initiatives, may also offer valuable strategies for enhancing work ability across different healthcare professions.

## 5. Conclusions

This study reinforced the critical relationship between demographic and occupational factors and the Work Ability Index (WAI) among healthcare professionals. The findings confirmed that work ability declined with age, with older workers exhibiting lower WAI scores than their younger counterparts. Additionally, women workers reported lower WAI scores compared with men, highlighting potential gender-related disparities in occupational health. Furthermore, occupational roles significantly influenced work ability, with nurses, healthcare workers (HCWs), and midwives displaying the lowest WAI scores, likely due to the physically and mentally demanding nature of their roles.

However, this study was not without limitations. The cross-sectional design prevented the establishment of causal relationships, necessitating further longitudinal research to track changes in WAI over time and assess the long-term impact of aging and occupational demands on healthcare workers. Additionally, self-reported data may introduce bias as individual perceptions of work ability and health conditions can vary. Future studies should incorporate objective physiological and psychological assessments to complement self-reported measures. Furthermore, external factors such as organizational policies, workplace conditions, and psychosocial stressors were not extensively analyzed, warranting further exploration to understand their role in shaping work ability outcomes.

Future research should focus on intervention strategies aimed at improving work ability across different age groups and professional categories. Implementing targeted workplace modifications, ergonomic interventions, and wellness programs can help mitigate the negative effects of aging on work ability. Investigating flexible work arrangements, job rotation policies, and mental health support services can also contribute to a healthier and more sustainable workforce.

Ultimately, these findings underscore the urgent need for tailored occupational health strategies to support an aging workforce and address gender disparities in work ability. By adopting a proactive approach, healthcare institutions can enhance worker well-being, sustain productivity, and ensure the long-term resilience of the healthcare sector. As the healthcare industry continues to evolve, prioritizing workforce adaptability and occupational health will be key to maintaining high-quality patient care and a robust healthcare system.

### Limits

This study provides valuable insights into the factors influencing the Work Ability Index (WAI) among healthcare professionals, particularly in relation to gender, age, and occupational roles. However, several limitations should be considered when interpreting the findings.

One key limitation is the cross-sectional design of the study, which restricted our ability to establish causality between the variables examined. While the data suggested significant associations between WAI scores and demographic as well as occupational factors, these relationships should be interpreted with caution as cross-sectional studies capture only a snapshot in time. Consequently, it remains unclear whether the observed differences in WAI scores were the result of long-term trends or short-term influences.

A longitudinal approach would be necessary to determine whether WAI declines progressively with age and occupational demands or if fluctuations in work ability are linked to transient external factors such as workload variations, organizational changes, or short-term health conditions.

Another limitation is the potential for self-reporting bias. The WAI is based on self-assessments, which may be influenced by subjective perceptions, recall bias, or social desirability bias. For example, some participants may have overestimated or underestimated their work ability due to personal attitudes, job satisfaction, or external pressures.

Incorporating objective physiological and cognitive assessments, such as physical endurance tests, stress biomarkers, or neuropsychological evaluations, could provide a more comprehensive understanding of work ability and mitigate the influence of subjective reporting biases.

Additionally, the study was conducted within a single hospital setting in central Italy, which may limit the generalizability of the findings to other healthcare environments, different regions, or other occupational groups. Future research should consider multi-center or longitudinal designs to explore how work ability evolves over time and whether targeted interventions could mitigate age-related declines in work ability.

Expanding the study to include healthcare professionals from multiple hospitals, in both urban and rural areas, as well as from private and public healthcare settings, would enhance the external validity of the findings. Furthermore, cross-national comparisons could provide insights into how different healthcare systems influence work ability outcomes.

Finally, while this study controlled for demographic and occupational variables, confounding factors such as work environment conditions, psychosocial stressors, and specific job demands were not extensively analyzed. Future studies should incorporate a more detailed assessment of these factors to better understand their role in shaping WAI scores.

For instance, variables such as shift patterns, workload intensity, access to workplace support, and exposure to ergonomic risks could have a substantial impact on work ability. Investigating these elements through qualitative research or mixed-method approaches could offer a more nuanced perspective on the challenges faced by different healthcare professions.

## Figures and Tables

**Figure 1 healthcare-13-00702-f001:**
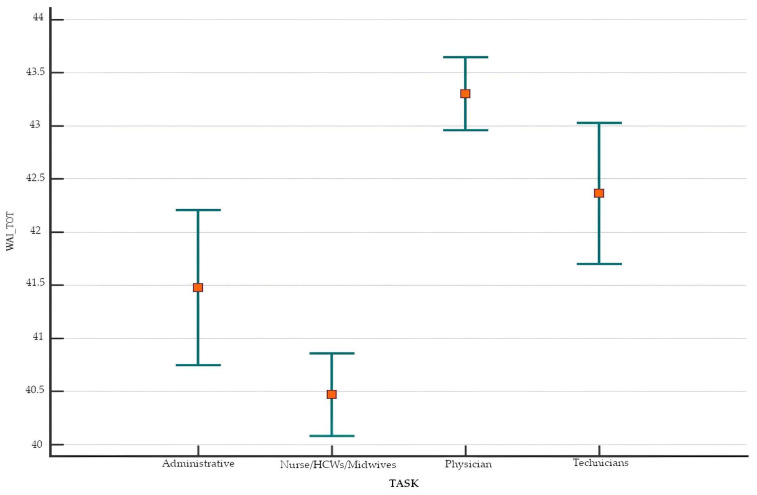
Means and confidence intervals of WAI by profession.

**Figure 2 healthcare-13-00702-f002:**
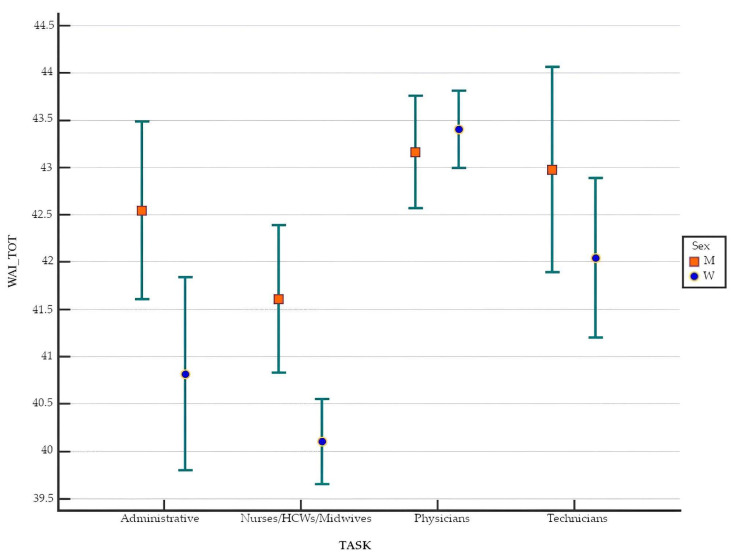
Means and confidence intervals of WAI by profession and gender.

**Figure 3 healthcare-13-00702-f003:**
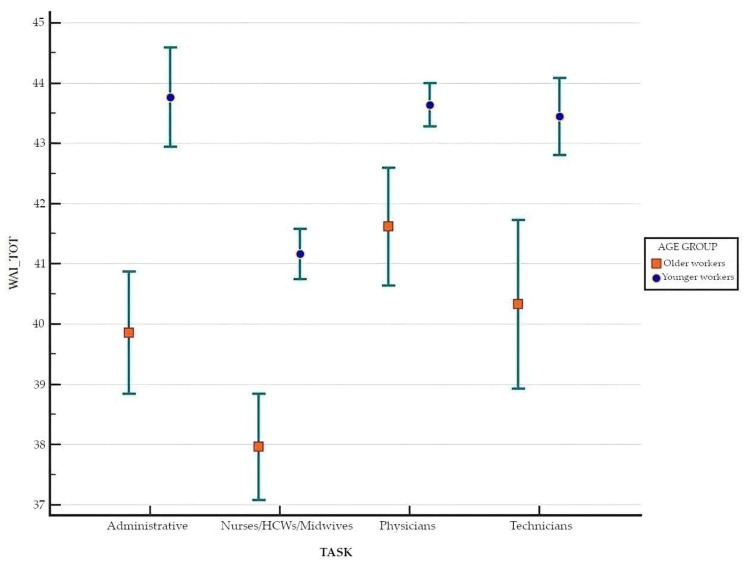
Means, standard deviations, and confidence intervals of WAI by profession and age.

**Table 1 healthcare-13-00702-t001:** Description of the sample under study with the percentage of gender, age groups, and profession.

Variables	n	%
Gender		
Women	1248	67.6
Men	599	32.4
Age group (yrs.)		
<55	1373	74.3
≥55	474	25.7
Profession		
Administrative personnel	207	11.2
Nurses/HCWs/midwives	882	47.8
Physicians	544	29.5
Technicians	214	11.6

**Table 2 healthcare-13-00702-t002:** WAI and its seven dimensions by gender and age group.

	Gender					Class of Age				
	M		F		*p*	<55 Y.O.		≥55 Y.O.		*p*
	Median	IQR	Median	IQR		Median	IQR	Median	IQR	
WAI	43.50	5.50	42.00	7.00	<0.001 ^a^	43.00	5.50	40.00	7.50	<0.001 ^a^
W1—Current work ability compared to highest work ability ever	8.00	2.00	8.00	1.00	0.168 ^a^	8.00	2.00	8.00	2.00	0.529 ^a^
W2—Work ability in relation to the demands of the job	9.00	2.00	9.00	2.00	0.002 ^a^	9.00	2.00	9.00	2.00	0.193 ^a^
W3—Number of current diseases diagnosed by a physician	5.00	4.00	5.00	5.00	<0.001 ^a^	5.00	3.00	3.00	4.00	<0.001 ^a^
W4—Estimated work impairment due to diseases	N	%	N	%	*p*	N	%	N	%	*p*
In my opinion I am entirely unable to work.	2	0.3%	2	0.2%	0.002 ^b^	1	0.1%	3	0.6%	<0.001 ^b^
Because of my condition. I feel I am able to do only part time work	0	0.0%	13	1.0%		4	0.3%	9	1.9%	
I must often slow down my work pace or change my work methods.	11	1.8%	35	2.8%		24	1.7%	22	4.6%	
I must sometimes slow down my work pace or change my work methods	30	5.0%	76	6.1%		59	4.3%	47	9.9%	
I am able to do my job. but it causes some symptoms.	63	10.6%	194	15.5%		146	10.6%	111	23.4%	
There is no hindrance	490	82.2%	928	74.4%		1138	82.9%	282	59.5%	
W5—Sick leave during the past 12 months	N	%	N	%	*p*	N	%	N	%	*p*
100–354 days	7	1.2%	15	1.2%	0.087 ^b^	11	0.8%	11	2.3%	<0.001 ^b^
25–99 days	27	4.5%	81	6.5%		62	4.5%	46	9.7%	
10–24 days	65	10.9%	154	12.4%		156	11.4%	63	13.3%	
Max 9 days	123	20.6%	291	23.4%		305	22.3%	109	23.0%	
None	374	62.8%	704	56.5%		835	61.0%	245	51.7%	
W6—Personal prognosis of work ability 2 years from now	N	%	N	%	*p*	N	%	N	%	*p*
Unlikely	16	2.7%	36	2.9%	0.017 ^b^	30	2.2%	22	4.7%	<0.001 ^b^
Not certain	33	5.5%	122	9.8%		84	6.1%	71	15.0%	
Relatively certain	548	91.8%	1089	87.3%		1259	91.7%	380	80.3%	
W7—Mental capacities	Median	IQR	Median	IQR	*p*	Median	IQR	Median	IQR	*p*
Mental capacities	4.00	1.00	4.00	1.00	<0.001 ^a^	4.00	1.00	4.00	1.00	0.002 ^a^

^a^ Mann–Whitney U test. ^b^ Chi-Square test.

**Table 3 healthcare-13-00702-t003:** WAI dimensions by profession.

	Profession 1:		Profession 2:		Profession 3:		Profession 4:		*p*
	Administrative		Nurses/HCWs/ Midwives		Physicians		Technicians		
	Median	IQR	Median	IQR	Median	IQR	Median	IQR	
WAI	42.50	6.00	42.00	6.75	44.00	5.00	44.00	6.00	<0.001 ^a^
W1—Current work ability compared to highest work ability ever	9.00	2.00	8.00	2.00	8.00	3.00	9.00	2.00	<0.001 ^a^
W2—Work ability in relation to the demands of the job	9.50	2.00	9.00	2.00	9.00	2.00	9.00	2.00	<0.001 ^a^
W3—Number of current diseases diagnosed by a physician	4.00	3.00	4.00	5.00	5.00	3.00	5.00	4.00	<0.001 ^a^
W4—Estimated work impairment due to diseases	N	%	N	%	N	%	N	%	*p*
In my opinion I am entirely unable to work.	0	0.0%	2	0.2%	0	0.0%	2	0.9%	<0.001 ^b^
Because of my condition. I feel I am able to do only part time work	1	0.5%	11	1.2%	0	0.0%	1	0.5%	
I must often slow down my work pace or change my work methods.	6	2.9%	34	3.9%	2	0.4%	4	1.9%	
I must sometimes slow down my work pace or change my work methods	14	6.8%	70	7.9%	12	2.2%	10	4.7%	
I am able to do my job. but it causes some symptoms.	32	15.5%	153	17.4%	50	9.2%	22	10.3%	
There is no hindrance	154	74.4%	611	69.4%	480	88.2%	175	81.8%	
W5—Sick leave during the past 12 months	N	%	N	%	N	%	N	%	*p*
100–354 days	4	1.9%	13	1.5%	1	0.2%	4	1.9%	<0.001 ^b^
25–99 days	18	8.7%	59	6.7%	16	2.9%	15	7.1%	
10–24 days	21	10.1%	134	15.2%	42	7.7%	22	10.4%	
Max 9 days	44	21.3%	238	27.0%	85	15.6%	47	22.2%	
None	120	58.0%	436	49.5%	400	73.5%	124	58.5%	
W6—Personal prognosis of work ability 2 years from now	N	%	N	%	N	%	N	%	*p*
Unlikely	5	2.4%	37	4.2%	7	1.3%	3	1.4%	<0.001 ^b^
Not certain	11	5.3%	110	12.5%	20	3.7%	15	7.0%	
Relatively certain	190	92.2%	735	83.3%	517	95.0%	196	91.6%	
W7—Mental capacities	Median	IQR	Median	IQR	Median	IQR	Median	IQR	*p*
Mental capacities	4.00	1.00	4.00	1.00	4.00	1.00	4.00	1.00	0.001 ^a^

^a^ Kruskal–Wallis test. ^b^ Chi-Square test.

## Data Availability

The data presented in this study are available on request from the corresponding author. The data are not publicly available due to privacy restrictions.

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
