# Peer review of "Correlations Between Gender, Age, and Occupational Factors on the Work Ability Index Among Healthcare Professionals"

_healthcare, 2025, doi:10.3390/healthcare13070702_

Round 1
Reviewer 1 Report
Comments and Suggestions for Authors
The study addresses a significant issue in occupational health, particularly relevant to the healthcare sector, where work ability can significantly impact both employee well-being and patient care.
- It may be beneficial to include a mention of study limitations within the abstract; for example, the cross-sectional nature of the study restricts causal inferences.
- Create a more fluid and engaging introduction that effectively presents the problem
- While validated questionnaires are mentioned, there is no detailed description of what these questionnaires entail—specifically which ones were used for WAI and metabolic syndrome assessment.
- Specify the inclusion or exclusion criteria for participant selection, which is important for transparency regarding the selection process.
- Correct the Conclusions. The conclusion could benefit from mentioning potential areas for future research that could build on the findings, thereby guiding subsequent studies in this field. Add also limitations of the study. Conclude with a powerful statement that encapsulates the importance of the findings and their implications for the future of occupational health in the healthcare sector.
- Some sentences are long and complex, which may hinder comprehension. Breaking them down into shorter, simpler sentences could improve clarity.
Reviewer 2 Report
Comments and Suggestions for Authors
Dear Author(s),
The manuscript entitled ‘‘Exploring the Impact of Gender, Age, and Occupational Factors on the Work Ability Index among Healthcare Professionals’’. I congratulate you all on the exciting manuscript. I think that the research topic is important in the field of occupational health and safety. However, this topic has been researched for years by different researchers, in different locations and with different target groups. Of course, the manuscript will contribute to the literature. However, considering the quality of the journal, I cannot accept a manuscript that contains many major problems. The problems are presented below.
Introduction
The introduction does not provide a background for the importance of the research. It seems to be just a collection of information to make some points. There is no coherence, and there are problems with the transition between paragraphs. While the title of the manuscript mentions the critical factor of gender, the introduction does not mention the advantages and disadvantages of gender differences in the workplace. Only the differences in index scores between men and women are mentioned. The introduction mentions the importance of an index that has been in use since 1981, but lacks information on how this research will fill the gap in the field. Frankly, it is not possible to understand where the issue will lead. Why is the Work Ability Index important? Which aspects have been presented so far, and which aspect will you address with this study? You should emphasize the originality of your research with the answers to these questions.
Materials and methods
The methodology section is confusing and does not provide the necessary content in the correct order. First, information about the research design is given. Then, there is information about the population and sample, including the participants. Subsequently, the data collection tools used, the procedure for working with the participants, and the variables should be included. In addition, research questions and hypotheses should be clearly stated. I would also like to emphasize that the number of participants in the research is 1847, which is a perfect number. On the other hand, Line 146- “Design of the study” could have been used instead of the title “Data.”
Line 154- “Study cohort” could have been replaced by “participants” or “population of the study”.
The “Wilcoxon” test is mentioned in the statistical analysis section, but this test is the nonparametric equivalent of the t-test for dependent groups. Dependent groups were never mentioned in the study. As far as I understand, participants were not given repeated measures.
Results
It seems to be well written but there are problems at a few points. For example, Nonparametric test results were written, and therefore, the research data were not normally distributed. However, giving mean ± SD in non-normally distributed data is not appropriate. Min-max and median values should be provided for non-normally distributed data. In general, there are problems in spelling. For example, in Line 270, “....... Chi Quadrato.......” the names are not written correctly.
The chi-square test result in Table 1 shows that the p-value is significant. However, it is not written which variable causes this difference. Therefore, the post hoc Chi-Square test was not performed. The same is true for the other tables.
Discussion
The research results are not sufficiently discussed according to the literature, and the information is written by rote.
Reviewer 3 Report
Comments and Suggestions for Authors
Thank you for inviting me to review this paper. This study examines the associations of gender, age, and occupational groups with WAI. The topic is both interesting and relevant to occupational health practitioners and researchers. However, I have several concerns regarding the manuscript before recommending it for publication.
Title
The assertion of causal impacts ("Exploring the impact") should be avoided in a cross-sectional study. It should be replaced with "correlations" or "associations."
Introduction
The introduction overly focuses on the effects of age on WAI, neglecting how gender or occupational groups are correlated with WAI. The authors should provide a more balanced discussion covering all key variables.
Materials and Methods
This section lacks a description of the main variable, WAI. The authors should explain the following:
(1) The items of WAI and its scoring system in a concise manner.
(2) The reliability and validity of WAI as reported in previous studies.
The authors employed both parametric and non-parametric tests to compare WAI across various groups, based on the K-S test. While from a purely statistical perspective, this may not be a flawed approach, normality tests such as the K-S test are overly sensitive in large samples. The authors should consider examining histograms or QQ plots to assess whether the distribution of WAI significantly deviates from normality. Moreover, dividing the analysis of a single variable, such as WAI, into both parametric and non-parametric methods within a single study may compromise the consistency of the manuscript. I recommend using t-tests or ANOVA consistently throughout the manuscript, as the authors presented their data using the mean and SE (rather than the median and IQR), which aligns with parametric tests. If the authors prefer to use non-parametric tests, the mean and SE should be replaced with the median and IQR.
Results
-
- A descriptive table of the sample characteristics should be included.
- Figure titles should be corrected to proper English.
- Figure 1 does not present the results of post-hoc analyses; it merely presents the mean and SE of the WAI values. The same applies to Figures 2 and 3.
- Figure 3 is mislabeled: young workers should be shown in blue, and older workers in orange
Discussion
-
-
- Lines 284–287: The authors state in the first sentence that numerous studies "consistently" demonstrate an inverse correlation between age and WAI, but in the following sentence, they maintain that there is "a lack of consensus" in the literature. This transition is abrupt. The discussion should expand on how previous studies show inconsistency in the association between age and WAI.
- Lines 297–299: Please add a reference to support this statement.
- Lines 300–308: References should be added to this paragraph.
- Lines 319–324: References should be added to these sentences.
- Lines 338–348: References should be added to this paragraph.
- The potential limitations of this study, such as its cross-sectional design, should be elaborated upon in the discussion section.
-
Conclusion
-
- The use of the term "confirms" is not appropriate for this study. The authors observed only "correlations" between variables. A more measured tone should be adopted to avoid making definitive claims.
- Throughout the manuscript, the authors should take greater care to avoid causal language (e.g., impact, effect).
Round 2
Reviewer 1 Report
Comments and Suggestions for Authors
The paper is improved and can be published. I just suggest to use the title Limitations of the study, not Limits.
Author Response
“The paper is improved and can be published. I just suggest to use the title Limitations of the study, not Limits.”
Reply: Thank you for your feedback and for recognizing the improvements in our paper. We appreciate your suggestion and have updated the section title to "Limitations of the study" accordingly.
Reviewer 2 Report
Comments and Suggestions for Authors
Dear Author(s),
I appreciate your effort in making the necessary revisions, but I still think that the manuscript is not of sufficient quality to be published in this journal. The author's response to my suggestions is not clear. I still think that the manuscript has both general design and methodological shortcomings, which I list below.
- In the introduction you have tried to emphasize the importance of the manuscript in line with my suggestions. However, there is not a single citation between lines 40-61.
- Research hypotheses or questions were not included in the Method section or at the end of the Introduction.
- The large sample size is a means for the data to be normally distributed. Evidence could have been presented by looking at the skewness and kurtosis values as well as Kolmogorov Smirnov and Shapiro - Wilk tests to see whether the data were normally distributed. This issue frankly raises the question of whether mistakes were made in other tests. I suggest getting help from a statistician, because reaching 1847 people is a great achievement. Working with a statistician afterwards may not be very pleasant, but at least it will prevent making mistakes.
- Table 1 has been added, I think in line with the suggestions of other referees. However, the table is not in accordance with the rules of construction, the beginning of the table starts with “gender”. It should have started with “variables” instead.
Reviewer 3 Report
Comments and Suggestions for Authors
Thank you for addressing my comments during the revision process.
Author Response
“Thank you for addressing my comments during the revision process.”
Reply: We sincerely appreciate your thoughtful comments and feedback throughout the revision process.
Round 3
Reviewer 2 Report
Comments and Suggestions for Authors
Dear Author(s),
Thank you for making the revisions in line with my recommendations. However, I could not see the references and research questions in the introduction section, so it is suitable for publication after these two additions are made.
Author Response
“Thank you for making the revisions in line with my recommendations. However, I could not see the references and research questions in the introduction section, so it is suitable for publication after these two additions are made.”
Reply: We thank the reviewer for carefully considering our revisions and for providing such precise and insightful feedback. In the hope of making a significant improvement to our manuscript, we are submitting the following revisions.
- “ […] However, I could not see the references […]”
Reply: We thank the reviewer and apologize for this omission; the citations have been incorporated into the text as requested.
- “[…] I could not see the references and research questions in the introduction section. […]”
Reply: Thank you for your valuable feedback. In response to your comment regarding the absence of research hypotheses or questions at the end of the Introduction, we have now explicitly stated the research questions that guide this study at the conclusion of the Introduction section.